# Maverick: Personalized Edge-Assisted Federated Learning with Contrastive Training

## Abstract

In an edge-assisted federated learning (FL) system, edge servers aggregate the local models from the clients within their coverage areas to produce intermediate models for the production of the global model. This significantly reduces the communication overhead incurred during the FL process. To accelerate model convergence, FedEdge, the state-of-the-art edge-assisted FL system, trains clients' models in local federations when they wait for the global model in each training round. However, our investigation reveals that it drives the global model towards clients with excessive local training, causing model drifts that undermine model performance for other clients. To tackle this problem, this paper presents Maverick, a new edge-assisted FL system that mitigates model drifts by training personalized local models for clients through contrastive local training. It introduces a model-contrastive loss to facilitate personalized local federated training by driving clients' local models away from the global model and close to their corresponding intermediate models. In addition, Maverick includes anomalous models in contrastive local training as negative samples to accelerate the convergence of clients' local models. Extensive experiments are conducted on three widely-used models trained on three datasets to comprehensively evaluate the performance of Maverick. Compared to state-of-the-art edge-assisted FL systems, Maverick accelerates model convergence by up to 16.2x and improves model accuracy by up to 12.7%.

## Keywords

Edge-assisted federated learning, model drift, contrastive learning

### ACM Reference Format:
Anonymous Author(s). 2018. Maverick: Personalized Edge-Assisted Federated Learning with Contrastive Training. In *Proceedings of Make sure to enter the correct conference title from your rights confirmation emai (Conference acronym 'XX)*. ACM, New York, NY, USA, 11 pages. https://doi.org/XXXXXXX.XXXXXXX

## 1 Introduction

Edge devices, such as mobile and Web-of-Things (WoT) devices, account for over half of global Internet traffic and produce vast and varied data that fuel a wide variety of machine learning (ML) applications, e.g., recommender systems [62] and social networks [63]. The ML models powering these applications rely on private data

collected or produced by mobile and WoT devices, leading to significant privacy concerns. Federated learning (FL) [43] has recently attracted significant attention as an effective method for training ML models in a privacy-preserving manner across edge devices (often referred to as *clients*). It enables clients to collectively train a shared global ML model with the coordination of a cloud server. In each training round, clients independently train local models on their data and then transmit local models[1] to the cloud server. The cloud server aggregates these local models into a global model with a method like FedAvg [43] and then distributes the global model to clients for the next training round. The iterative process allows the cloud server to incorporate clients' knowledge into the global model while preserving users' data privacy.

Clients and the cloud server frequently exchange local models, incurring massive traffic overhead. This issue is further exacerbated by the increasing size of modern ML models, driven by the need for higher accuracy. Recently, researchers have started to utilize the benefits of edge computing to mitigate the traffic overhead generated by FL systems [17, 39, 54, 60]. Edge computing, a key 5G technology, decentralizes storage and computing resources by graphically placing edge servers closer to clients, such as regional data centers and base stations, reducing reliance on the central cloud. In an edge-assisted FL system, clients transfer local models to nearby edge servers, which aggregate these models into intermediate models and transfer them to the cloud. The cloud server aggregates these intermediate models for the production of a global model and transmits it to clients for the next global training round. Transmitting only intermediate models to the cloud server, edge-assisted FL significantly mitigates the traffic overhead [17, 54, 57].

In an edge-assisted FL system, model convergence can be accelerated by enabling clients to perform *local* federated training while awaiting the global model from the cloud server during each global training round [54]. As illustrated in Fig. 1, after transmitting the intermediate model to the cloud server, an edge server starts local FL by sending the intermediate model to its clients. These clients continue to train their local models with the coordination of the edge server until it receives the global model from the cloud server. Upon receiving the global model, the edge server stops the local federated training, aggregates the global model with the intermediate model, and sends the updated intermediate model to its clients for the next global training round.

Local federated training accelerates model convergence on independently and identically distributed (IID) data by allowing clients to leverage the waiting time for the global model from the cloud server. This is verified in Fig. 3 and Fig. 4. However, in real-world FL systems, clients' data are often non-IID. A straggler client in the system can easily lead to *model drifts* on non-IID data. Clients in an

---

[1] In fact, clients transmit model updates rather than models to the cloud in an FL system. In this paper, we speak of them interchangeably for ease of exposition.

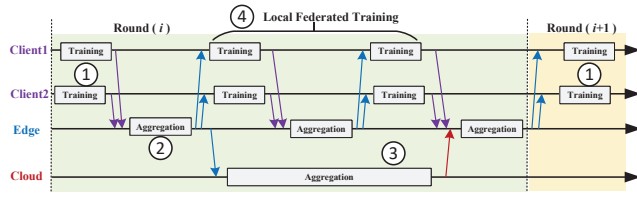

**Figure 1: Edge-assisted FL with local federated training. Each global training round consists of four steps: ① Client Training, ② Edge Aggregation, ③ Cloud Aggregation, and ④ Local Federated Training.**

edge-assisted FL system often vary in terms of computational capabilities, communication bandwidth, and energy resources [1, 44, 55]. As a result, clients' local models arrive at their corresponding edge servers at different times. Both edge servers and the cloud server must wait for all incoming models before they can perform model aggregation. A straggler client in the system can easily increase other clients' waiting time for the global model [5, 7, 48]. These clients perform more local federated training rounds than others. Our experimental investigation revealed that this drives the global model toward these clients and causes a model drift. As demonstrated in Fig. 5, clients who perform more local federated training rounds achieve a higher model accuracy at the price of a decreased model accuracy for other clients. This comes with undesirable consequences. For example, clients with limited resources, knowing that they will be disadvantaged, may be reluctant to participate in the FL system.

A straightforward solution to model drifts is to remove straggler clients from the system [30, 33, 47]. For example, FedCS [47] selects the best clients based on their computational power and network conditions for FL. This minimizes the difference in the number of clients' local federated training rounds. However, excluding straggler clients can lead to the loss of valuable knowledge, potentially causing a reduction in model accuracy. Oort [30] and PyramidFL [33] optimize client selection based on statistical and system efficiency to improve model accuracy and accelerate training. However, they cannot exclude knowledgeable clients with limited resources. As a result, some clients still perform more local federated training than others, causing model drifts. In addition, these methods favor high-performing clients. Clients with limited resources risk being excluded from the learning system. This is unfair to them.

This paper introduces Maverick, a new edge-assisted FL system that aims to mitigate the model drift caused by imbalanced local training. Maverick trains personalized models for individual clients through contrastive learning, guiding clients' local models away from the global model and close to their corresponding intermediate models. In addition, clients' models often differ in their quality, due to the quality and quantity of their training data [30, 33], as well as potential threats like data poisoning attacks [2, 23] and model poisoning attacks [14, 24]. Poor-quality local models (referred to as *anomalous models* hereafter) can compromise the quality of intermediate models, as they have different even opposite convergence directions to the global optima. Inspired by contrastive learning that negative samples are essential in guiding model training, unlike

existing FL systems [14, 54] that employ defense mechanisms to exclude anomalous models, Maverick leverages anomalous models as negative samples, driving the training of clients' models in the right direction away these anomalous models. This paper's key contributions include:

- To the best of our knowledge, Maverick is the first edge-assisted FL system that mitigates the model drift issue caused by clients' imbalanced local training. It alleviates clients' concerns about being disadvantaged in the FL system.
- Maverick introduces a personalized model-contrastive loss to help clients train personalized local models, effectively mitigating model drifts in edge-assisted FL systems (§5.1).
- Maverick introduces an anomalous model-contrastive loss in clients' local model training, leveraging anomalous models as negative samples to accelerate model convergence and improve model accuracy (§5.2).
- Extensive experiments are performed on three widely-used public datasets with three ML models. The results show that Maverick outperforms the state-of-the-art edge-assisted FL system, increasing model accuracy by 5.2%-12.7% and speeding up convergence by 1.4x-16.2x.

## 2 Background

**Edge-assisted FL.** In an edge-assisted FL system [39, 54, 60], a group of clients $U = \{u_1, \ldots\}$ is served by a set of $M$ edge servers $S = \{s_1, \ldots, s_M\}$ and the cloud server $C$. Each edge server $s_m \in S$ manages a subset of clients, and each client $u$ trains a local model $W_L$ on its dataset $\mathcal{D}_u = \{(x_i, y_i)\}_{j=1}^{|\mathcal{D}_u|}$. Fig. 1 shows the overview of an edge-assisted FL system that involves local federated training. Each global training round goes through 4 steps. ① Client Training: Clients train local models on their data and send them to their edge servers. ② Edge Aggregation: Edge servers aggregate clients' local models for the production of intermediate models. These intermediate models are then transmitted to the cloud for the production of the global model and distributed to their clients for local federated training. ③ Cloud Aggregation: After the cloud server receives intermediate models from edge servers, it aggregates these models into a global model and then distributes it back to the edge servers. ④ Local Federated Training: Clients update their local models to intermediate models and train local models on their datasets. Next, clients transfer local models to their edge servers. If an edge server receives the global model from the cloud server, it aggregates the global model with its intermediate model to generate a new intermediate model. Then it sends the updated intermediate model to its clients for the next global training round. Otherwise, it sends the intermediate model to clients for the next local federated training. Steps ③ and ④ are performed in parallel. In each global training round, each set of clients may perform zero or many rounds of local federated training.

**Contrastive Learning.** Contrastive learning aims to learn an embedding space that guides similar data points closer while forcing dissimilar ones farther apart. It achieves this goal by minimizing the distance between the positive samples and maximizing the distance between negative samples in that space. For example, when training an image classification model, a contrastive loss is introduced to maximize the similarity between differently augmented views of

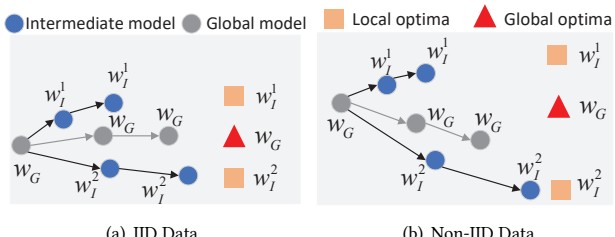

(a) IID Data      (b) Non-IID Data

**Figure 2: Illustration of model convergence direction in edge-assisted FL system under both IID and non-IID data, where edge server $s_2$ and its clients perform more local federated training.**

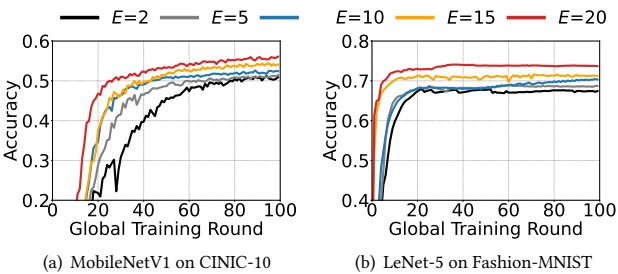

(a) MobileNetV1 on CINIC-10      (b) LeNet-5 on Fashion-MNIST

**Figure 3: Comparison of global model convergence and accuracy across varying numbers of local federated training in edge-assisted FL system under IID data, where $E = 5$ denotes one of the edge servers and its clients perform 5 local federated training rounds, while the other edge server and its clients conduct 2 only local federated training rounds in each global training round.**

the same image [8]:

$$l_{i,j} = -\log \frac{\exp(sim(x_i, x_j)/\tau)}{\sum_{k=1}^{2N} \exp(sim(x_k, x_i)/\tau)} \quad (1)$$

where $x_i$ and $x_j$ represent two distinct views of the same image, $N$ is the number of images, $sim(,)$ denotes the cosine similarity function, and $\tau$ is a temperature parameter. The loss is calculated by summing the contrastive loss over all image pairs. Unlike traditional contrastive learning that compares the representations of different images, Maverick performs contrastive local training by comparing different models, i.e., the global model, intermediate models, and anomalous models.

## 3 Motivation

In an edge-assisted FL system, clients perform local federated training when they wait for the global model. This accelerates the convergence of the global model. In each global training round, edge servers must wait for the last local model before they can perform model aggregation. However, clients often differ in their computing capacities and network conditions. The time required for each client to complete the training and transmission of its local model varies. As a result, they may perform different numbers of local federated training rounds in the same global training round. In particular, straggler clients can significantly amplify the imbalance in clients' local federated training. Fig. 2(a) shows the effect of the number of local federated training rounds on the convergence of a global model trained on IID data. The edge-assisted FL system

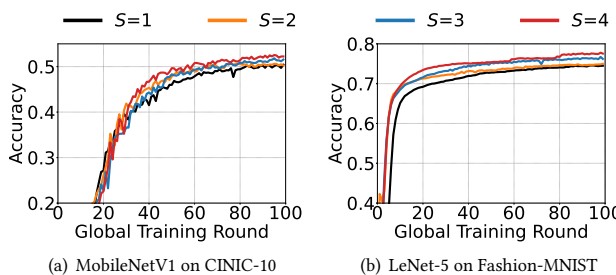

(a) MobileNetV1 on CINIC-10      (b) LeNet-5 on Fashion-MNIST

**Figure 4: Comparison of global model convergence and accuracy with different numbers of edge servers perform more local federated training in FedEdge under IID data, where $S = 3$ denotes three edge servers and their clients perform more local federated training (i.e., 5), while the remaining edge servers and their clients with stragglers only perform 2 local federated training in each global training round.**

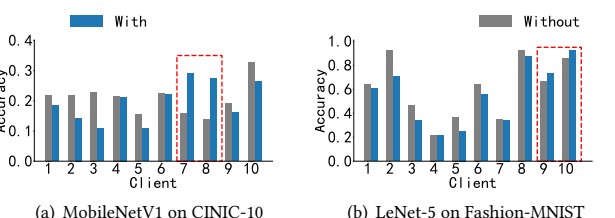

(a) MobileNetV1 on CINIC-10      (b) LeNet-5 on Fashion-MNIST

**Figure 5: Comparison of global model accuracy on clients' data in edge-assisted FL system under non-IID settings. "With" denotes that the edge servers covering $u_7$ and $u_8$ in CINIC-10 and $u_9$ and $u_{10}$ in Fashion-MNIST perform more local federated training rounds than other edge servers. "Without" denotes that all edge servers perform an equal number of local training rounds.**

involves two edge servers, $s_1$ and $s_2$, each covering 5 clients, and there is one straggler client within $s_1$' coverage. As a result, $s_2$ and its clients perform more local federated training rounds than $s_1$ and its clients. As a result, $W_I^2$ accelerates global model convergence. This indicates that increased local federated training effectively speeds up global model convergence under IID data. We conducted another experiment that involves two edge servers, one and its clients with stragglers performing only 2 local federated training rounds per global training round, and the other performing a varying number of local federated training rounds, i.e., 2, 5, 10, 15, and 20, with its clients. Fig. 3 compares the global model convergence and accuracy. We can see that an increased number of local federated training rounds from 2 to 20 accelerates the convergence of the global model and improves its accuracy. In addition, Fig. 4 compares global model convergence and accuracy across 5 edge servers under IID data. It shows the impact of the number of edge servers and their clients performing more local federated training rounds. These results show that local federated training can indeed accelerate model convergence and improve model accuracy under IID data.

However, clients' data are often non-IID in real-world applications [34, 37, 67]. We found that imbalanced local federated training results in global model drift under non-IID settings. Fig. 2(b) shows the effect of local federated training on global model convergence under non-IID settings, where $s_2$ and its clients perform more local

**Figure 6: Maverick overview. It consists of supervised learning loss, personalized model-contrastive loss, and anomalous model-contrastive loss to tackle model drifts and accelerate model training. Edge servers are used to filter out anomalous models, aggregate local models as intermediate models, and receive the global model from the cloud server. This example only shows a client, an edge server, and the cloud server. In reality, multiple edge servers are connected to the cloud server, and multiple clients are within the coverage of each edge server.**

federated training rounds than $s_1$ and its clients. We can see that the global model aligns closer with $W_I^2$ than $W_I^1$. This demonstrates that the global model achieves a higher accuracy for clients with more local federated training rounds. Fig. 5 shows the results of two experiments that compare the accuracy of the global model, MobileNetV1 and LeNet-5, on different clients' data. Both experiments involve 5 edge servers, each covering two clients, and each client's data is restricted to a single class to set up a non-IID configuration. Specifically, in Fig. 5(a), $u_7$ (with label 7) and $u_8$ (with label 8) are covered by same edge server, performing more local federated training rounds than others. Similarly, in Fig.5(b), client $u_9$ (with label 9) and client $u_{10}$ (with label 10) are covered by the same edge server and perform more local federated training than other clients. We can see that the global model shows improved accuracy on $u_7$, $u_8$, $u_9$, and $u_{10}$ in their respective experiments. In addition, the average accuracy achieved across all clients drops by 1.1% for MobileNetV1 and 5.0% for LeNet-5. The results tell us that imbalanced local federated training can lead to global model drift under non-IID settings. When clients update the drifted global model to their local models, their local models may diverge significantly from their local objectives. This divergence can discourage clients from continuing their participation in the FL system.

## 4 Maverick Overview

Maverick is a new edge-assisted FL system designed to mitigate model drifts. Fig. 6 presents an overview of Maverick. In each global training round, Maverick goes through 6 steps. ① Client Training: Clients train their local models by incorporating supervised learning loss, personalized model-contrastive loss, and anomalous model-contrastive loss. ② Client Transmission: Clients transmit local models to their edge servers. ③ Edge Aggregation: Upon receiving clients' local models, each edge server categorizes these models into two sets: a set of genuine models and a set of anomalous models. Next, it aggregates the genuine models to produce an intermediate model. ④ Edge Distribution: Each edge server distributes its intermediate model and anomalous models to its

clients. ⑤ Edge Transmission: Each edge server also transmits the intermediate model to the cloud server. ⑥ Cloud Aggregation and Distribution: The cloud server aggregates intermediate models from edge servers for the production of a global model, and then distributes it back to edge servers. Clients and edge servers repeatedly perform local federated training through Steps ① - ④ until the edge servers receive a global model from the cloud server. Upon receiving the global model, each edge server performs Step ③ to produce an intermediate model and anomalous models. The intermediate model is then aggregated with the global model to produce an updated intermediate model. Finally, the edge server distributes the global model, the intermediate model, and anomalous models to its clients. When clients receive these models, they start to perform the next global training round.

## 5 Maverick Training

To tackle the model drift issue caused by imbalanced local federated training in edge-assisted FL systems, Maverick introduces a personalized model-contrastive loss to facilitate local training by guiding clients' local models towards corresponding intermediate models while away from the drifted global model (§5.1). To further accelerate model convergence and improve model accuracy, Maverick incorporates an anomalous model-contrastive loss in clients' local federated training by driving clients' local models away from anomalous models (§5.2).

## 5.1 Personalized Model-Contrastive Training

Existing edge-assisted FL systems like FedEdge [54] drive the global model toward clients with more local federated training, causing model drifts that undermine the model performance on other clients' data. Recently, contrastive learning [8] has gained prominence as a method for training ML models using unlabeled data. Its key concept is to minimize the distance between positive samples while maximizing the distance from negative samples. Moon [34] was the first to utilize contrastive learning in FL, with the goal of improving global model performance on non-IID data. Moon tries

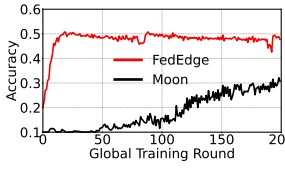

(a) MobileNetV1 on CINIC-10

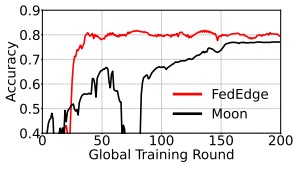

(b) LeNet-5 on Fashion-MNIST

Figure 7: Comparison in model convergence and accuracy between pure FedEdge [54] and FedEdge combined with Moon [34].

to move client models closer to the global model and away from their previous versions, guiding the global model toward the global optimum.

At the first glance, Moon seems capable of mitigating the global model drifts by aligning local models with the global model. Fig. 7 compares the convergence and accuracy of clients' models on their data between FedEdge and FedEdge combined with Moon. The results show that Moon actually slows down the convergence and decreases the accuracy of the model trained with FedEdge. For example, Fig.7(b) shows that FedEdge manages to converge the LeNet-5 model [31] to 77.1% accuracy on Fashion-MNIST [61] with 36 training rounds, while FedEdge combined with Moon requires 194 rounds to reach the same accuracy. Moreover, FedEdge achieves a final accuracy of 81.82%, while FedEdge combined with Moon only makes 77.1%. The reason is that Moon forces clients' local models to align with the drifted global model. This exacerbates the model drift issue.

Maverick addresses this issue by aligning clients' local models to their corresponding intermediate models rather than the global model. Specifically, when training their local models, clients regard the intermediate models transmitted from edge servers as positive samples and the global model as a negative sample. In this way, Maverick drives their local models to align with the intermediate models and away from the global model.

During a local federated training round, client $u$ receives the global model $W_G$ and the intermediate model $W_I$ from its edge server. For each input $x$, client $u$ extracts its representation $z_G = R_{W_G}(x)$, $z_I = R_{W_I}(x)$ and $z_L = R_{W_L}(x)$ from $W_G$, $W_I$ and $W_L$, respectively. Maverick's objective is to minimize the distance between $z_L$ and $z_I$ while maximizing the distance between $z_L$ and $z_G$ with a personalized model-contrastive loss:

$$l_p = -\log \frac{\exp(sim(z_L, z_I)/\tau)}{\exp(sim(z_L, z_I)/\tau) + \exp(sim(z_L, z_G)/\tau)} \quad (2)$$

where $\tau$ is a temperature parameter, and $sim(,)$ is the cosine similarity function.

## 5.2 Anomalous Model-Contrastive Training

In an edge-assisted FL system, the quality of clients' local models can vary significantly due to factors such as the size and quality of their training data. These models negatively impact the convergence and accuracy of the intermediate models [12, 25, 54]. In addition, adversarial clients may transmit poisoned models to edge servers, aiming to compromise the training process [14, 52, 59]. Recently, many methods have been proposed to detect and filter anomalous models in FL systems before the cloud server and edge

servers aggregate models [3, 13, 16, 28, 54]. In contrastive learning, negative samples are essential in guiding model training [4, 8, 9] by driving the model towards positive samples and away from negative samples. In FL, anomalous models usually aim to deviate the global model away from the global optimum. Thus, anomalous models also provide valuable information for guiding model convergence in edge-assisted FL systems. Maverick introduces an anomalous model-contrastive loss during local federated training, where anomalous models are regarded as negative samples. Since model detection is not the focus of this paper, Maverick leverages existing detection methods [3, 14, 54] to distinguish between anomalous models and genuine models.

In Maverick, when an edge server receives local models from its clients, it categorizes these models into a set of genuine models and a set of anomalous models with Adaptive-Krum [54]. Then, it transmits the anomalous models to its clients. When a client $u$ receives these anomalous models, denoted by $A$, it can perform contrastive learning to train its local model. Specifically, $u$ runs every input $x$ through each of these anomalous models, to obtain the representations of $x$: $z_A = R_{W_A}(x)$, where $W_A \in A$. Maverick tries to minimize the distance between $z_L$ and $z_I$ while maximizing the distance between $z_L$ and all $z_A$. Unlike personalized model-contrastive training (§5.1), which only utilizes a single negative sample (i.e., $W_G$), Maverick incorporates multiple negative samples, specifically the anomalous models from $A$. A straightforward approach is to calculate the anomalous model-contrastive loss to average the contrastive loss between $W_L$ and all anomalous models:

$$l_a = -\sum_A \frac{1}{|A|} \log \left( \frac{\exp(sim(z_L, z_I)/\tau)}{\exp(sim(z_L, z_I)/\tau) + \exp(sim(z_L, z_A)/\tau)} \right) \quad (3)$$

**Anomalous Model-Contrastive Loss.** Eq. 3 assumes that anomalous models have the same effect on model convergence. In fact, anomalous models deviate from the global optimum to varying degrees. Their impacts on model convergence are different. Maverick introduces an anomalous model-contrastive loss based on the distances between each anomalous model and all genuine models. As shown in Eq. 4, $d_A$ represents the distance between $W_A$ and all genuine models, $d_T = \sum_A d_A$ denotes the total distance between all anomalous models and all genuine models. In this way, anomalous models further from genuine models contribute more significantly to ensuring the correct model convergence direction.

$$l_a = -\sum_A \frac{d_A}{d_T} \log \left( \frac{\exp(sim(z_L, z_I)/\tau)}{\exp(sim(z_L, z_I)/\tau) + \exp(sim(z_L, z_A)/\tau)} \right) \quad (4)$$

**Top-$k$ Selection Mechanism.** During anomalous model-contrastive training, clients receive anomalous models from edge servers and compute the contrastive loss. We found that including too many anomalous models in local training does not yield further improvements in model convergence. Therefore, Maverick leverages a top-$k$ selection mechanism, choosing the anomalous models that contribute the most effectively to local federated training. An edge server first categorizes local models into a set of anomalous models and a set of genuine models. Next, it computes a distance $d_A$ between each anomalous model and all genuine models. Finally, it selects the $k$ anomalous models with the largest distances for transmission to the clients.

## 5.3 Overall Training

In addition to contrastive learning losses, Maverick also includes the supervised learning loss below in clients' model training:

$$l_{sup} = \frac{1}{|\mathcal{D}_k|} \sum_{i=1}^{|\mathcal{D}_k|} l(W_L; x_i, y_i) \tag{5}$$

Combining this loss with the personalized model-contrastive loss (Eq. 2) and the anomalous model-contrastive loss (Eq. 4), the total training loss for clients' local models is computed as follows:

$$l = l_{sup} + \mu_p \cdot l_p + \mu_a \cdot l_a \tag{6}$$

where $\mu_p$ and $\mu_a$ are the hyper-parameters for controlling the contributions of personalized model-contrastive loss and anomalous model-contrastive loss, respectively. Maverick's extra computation and communication overheads are discussed in Appendix A.1. Its pseudocode can be found in Appendix A.2.

## 6 Evaluation

### 6.1 Experimental Setup

**Environment.** Maverick is performed in an edge-assisted FL system comprised of five physical machines acting as edge servers within a private data center, and 50 clients are distributed across these servers. The cloud server is hosted on an Amazon c5.2xlarge EC2 instance. The round-trip times (RTTs) between the cloud and clients vary from 150 to 300 milliseconds, while the RTTs between edge servers and clients vary from 10 and 40 milliseconds, closely aligning with typical latencies observed in commercial 5G networks.

**Models and Datasets.** We train the LeNet-5 model [31] the on Fashion-MNIST dataset [61], the ResNet-34 model [21] on the CIFAR-10 dataset [29], and the MobileNetV1 model [22] on the CINIC-10 dataset [11]. These models and datasets have been widely used in FL studies for their mobile-friendliness [49, 64, 66, 68]. They are implemented with Python v3.6.2 and Torch v1.10.2 and trained with Stochastic Gradient Descent[2]. We set the learning rate, momentum, and weight_decay are 0.01, 0.9, and $5e^{-4}$, respectively.

**Baselines.** Maverick is compared against the following representative baselines.

- **HybridFL** [60]. HybridFL is a traditional edge-assisted FL system. In this setup, clients send local models to edge servers, which aggregate the local models into intermediate models. Next, these intermediate models are transferred to the cloud server, then these models are further aggregated to form a global model for the next training round.
- **FedProx** [35]. The FedProx system follows the same training process as FedEdge and imposes a proximal term to drive clients' local models to align with the global model.
- **FedPVR** [32]. FedPVR is a state-of-the-art personalized FL system. In the FedPVR system, when training local models, clients share the general layers only and retain the classification layers to enable model personalization.
- **Moon** [34]. This FL system was introduced in Section §5.1. It drives clients' local models to align with the global model, similar to FedProx, but with a contrastive learning loss.

---

[2]The source code is available at Maverick

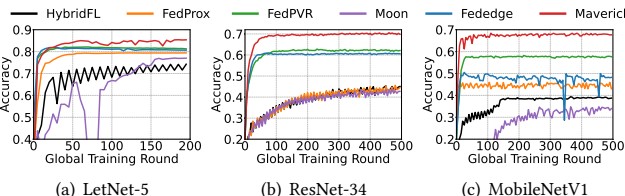

**Figure 8: Model convergence and accuracy across various methods, where LetNet-5, ResNet-34 and MobileNetV1 are trained on Fashion-MNIST, CIFAR-10 and CINIC-10.**

- **FedEdge** [54]. As the state-of-the-art edge-assisted FL system, FedEdge follows a similar process as HybridFL and incorporates local federated training to improve model convergence, as detailed in Section §2.

### 6.2 Overall Comparison

**Top-1 Accuracy.** Table 1 presents a comparison of Maverick's top-1 accuracy and baselines across three models on three datasets where FedProx, FedRep, and Moon also incorporate local federated training. The key observations are as follows: 1) Compared to HybridFL, other systems all achieve a higher top-1 accuracy, indicating that local federated training can indeed improve model convergence. 2) FedEdge outperforms FedProx and Moon. For example, on CIFAR-10 with MobileNetV1, FedEdge achieves an accuracy advantage of 18.8% and 1.2% over FedProx and Moon, respectively. This is attributed to the fact that FedProx and Moon drive clients' local models to align with the drifted global model. 3) FedPVR achieves accuracy improvements over FedEdge by 0.6%, 1.6%, and 4.0% across all models. This demonstrates that maintaining personalized classifiers locally mitigates model drifts, but only modestly. 4) Among all six systems, Maverick achieves the highest model accuracy, highlighting its effectiveness in mitigating model drifts in edge-assisted federated learning. Compared to FedEdge, the state of the art, Maverick achieves an accuracy advantage of 5.2%, 7.5%, and 12.7% across all models.

**Model Convergence.** Fig. 8 illustrates the convergence of the models trained in different systems, where LeNet-5, ResNet-34 and MobileNetV1 are trained on Fashion-MNIST, CIFAR-10, and CINIC-10, respectively. The results clearly demonstrate that Maverick achieves the greatest speedups in model convergence and as well as the highest accuracy. This observation aligns with the results presented in Table 1.

**Training Speedup.** Table 2 presents the training speedups achieved by Maverick over FedEdge across three datasets, ranging from 1.4x to 16.2x. Notably, on the CINIC-10 dataset, Maverick achieves a 14.9x speedup with the ResNet-34 model and a 16.2x speedup with the MobileNetV1 model. This underscores the significant impact of model drift on FedEdge's performance and highlights Maverick's superior ability to mitigate this issue.

### 6.3 In-Depth Evaluation

This section assesses Maverick's performance in various edge-assisted FL scenarios through a series of in-depth experiments.

Table 1: Full comparison of top-1 accuracy.

| Model | Dataset | Baseline | | | | | |
|---|---|---|---|---|---|---|---|
| | | HybridFL | FedProx | FedPVR | Moon | FedEdge | Maverick |
| LeNet-5 | Fashion-MNIST | 0.743 | 0.795 | 0.813 | 0.771 | 0.807 | **0.859** |
| ResNet-34 | CIFAR-10 | 0.393 | 0.456 | 0.628 | 0.440 | 0.610 | **0.707** |
| | CINIC-10 | 0.449 | 0.452 | 0.669 | 0.432 | 0.656 | **0.709** |
| MobileNetV1 | CIFAR-10 | 0.369 | 0.383 | 0.582 | 0.559 | 0.571 | **0.649** |
| | CINIC-10 | 0.392 | 0.427 | 0.578 | 0.354 | 0.508 | **0.684** |

Table 2: Maverick's speedup gains over FedEdge.

| Model | Dataset | Target Accuracy | Speedup over FedEdge |
|---|---|---|---|
| LeNet-5 | Fashion-MNIST | 0.813 | 4.1× |
| ResNet-34 | CIFAR-10 | 0.628 | 8.8× |
| | CINIC-10 | 0.669 | 14.9× |
| MobileNetV1 | CIFAR-10 | 0.571 | 1.4× |
| | CINIC-10 | 0.508 | 16.2× |

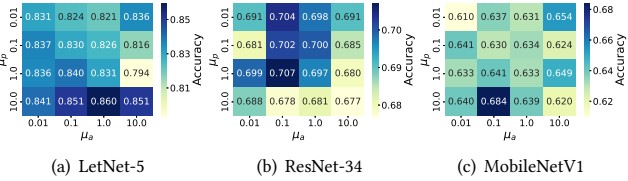

(a) LetNet-5          (b) ResNet-34          (c) MobileNetV1

Figure 9: Top-1 accuracy with different combinations of $\mu_p$ and $\mu_a$: a) LetNet-5 on Fashion-MNIST; b) ResNet-34 on CIFAR-10; c) MobileNetV1 on CINIC-10.

**Impact of $\mu_p$ and $\mu_a$.** Maverick includes two hyperparameters, $\mu_p$ and $\mu_a$, to adjust the contributions of personalized and anomalous model-contrastive losses, respectively, to clients' local model training. To evaluate their impacts on the performance of Maveric, we tune their values within {0.01, 0.1, 1, 10} and Fig. 9 shows the results. The optimal $\mu_p$ and $\mu_a$ of Maverick for LeNet-5, ResNet-34, and MobileNetV1 are {10, 1}, {1, 0.1}, and {10, 0.1}, respectively.

**Impact of Anomalous Model-Contrastive Loss ($l_a$).** To demonstrate the effect of the anomalous model-contrastive loss (§5.2), this experiment compares the model convergence and model accuracy when Maverick weights anomalous models differently and equally when clients train their local models. There are 5 edge servers in the system, each covering 10 clients. Under each edge server's coverage, 3 of the 10 clients have poor model quality implemented with model poison attacks [14]. Fig. 10 shows the results. Compared to weighs anomalous equally, Maverick weights anomalous models differently takes 62.8%, 79.7%, and 89.6% less time to converge the LeNet-5 model, the ResNet-34 model, and the MobileNetV1 model to 84.1%, 69.3%, and 64.9%, respectively. In addition, Maverick achieves an accuracy improvement of 1.1%, 1.4%, and 3.7%, respectively, for LetNet-5, ResNet-34, and MobileNetV1, when it weighs anomalous differently. This validates the discussion in Section §5.2 about anomalous models vs. genuine models, i.e., anomalous models more distanced from genuine models contribute more significantly to ensuring the correct model convergence direction.

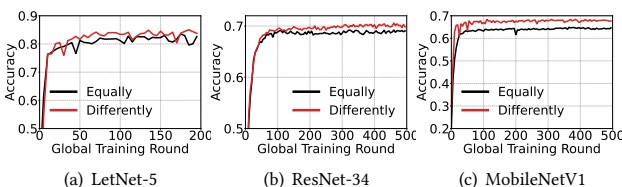

(a) LetNet-5          (b) ResNet-34          (c) MobileNetV1

Figure 10: Model convergence and accuracy when Maverick weighs anomalous models differently and equally: a) LetNet-5 on Fashion-MNIST; b) ResNet-34 on CIFAR-10; c) MobileNetV1 on CINIC-10.

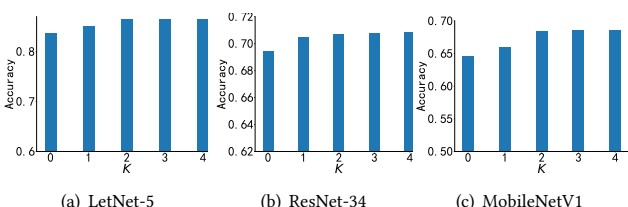

(a) LetNet-5          (b) ResNet-34          (c) MobileNetV1

Figure 11: Model accuracy with varying numbers of anomalous models included in the anomalous model-contrastive training: a) LetNet-5 on Fashion-MNIST; b) ResNet-34 on CIFAR-10; c) MobileNetV1 on CINIC-10.

**Impact of Number of Anomalous Models ($k$).** This experiment evaluates Maverick in model accuracy with varying numbers of anomalous models included in the anomalous model-contrastive training (§5.2). Fig. 11 presents the results, where $k$ anomalous models are selected based on their distance from genuine models. The results indicate an improvement in model accuracy as the initial increase in $k$ includes more anomalous models in clients' contrastive local training because Maverick can fuse diverse knowledge from both genuine models and anomalous models into clients' contrastive local training. However, as $k$ exceeds a certain threshold (e.g., 3 for LeNet-5, 2 for ResNet-34, and 3 for MobileNetV1), Maverick obtains no more significant accuracy gains. Thus, to optimize accuracy gains with minimal communication overhead, Maverick can include a small number of anomalous models, 3 in most cases, in clients' contrastive local training.

**Impact of Number of Local Federated Training ($E$).** This experiment compares model convergence and accuracy under Maverick with different numbers of local federated training rounds across three datasets. There are 5 edge servers in the FL system, each covering 10 clients. The number of local federated training varies from 2 to 20, performed by one edge server and its corresponding clients. As shown in Fig 12(c), when $E = 20$, Maverick converges to an accuracy of 71.5%, while the accuracy is 69.3%, 69.9% with $E = 2, 10$. Compared to the results in Section §3, FedEdge experiences accuracy degradation with an increase in local training.


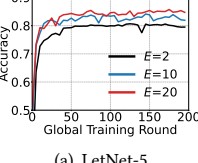
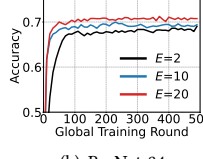
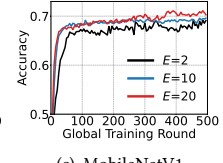

(a) LetNet-5      (b) ResNet-34      (c) MobileNetV1

**Figure 12: Model convergence and accuracy under Maverick with varying numbers of local federated training. Here, $E = 10$ indicates that one of the edge servers and its clients perform 10 local federated training within a single global round, while the others perform 2 local federated training rounds: a) LetNet-5 on Fashion-MNIST; b) ResNet-34 on CIFAR-10; c) MobileNetV1 on CINIC-10.**

In contrast, Maverick effectively mitigates the model drift issue inherent in FedEdge.

## 6.4 Ablation Study

Maverick includes personalized and anomalous model-contrastive loss during contrastive training. To assess the impact of the two components, ablation studies are conducted to evaluate each module under Maverick. Table 3 presents the model convergence accuracy with different model-contrastive losses: only personalized loss, only anomalous loss, and Maverick (including both losses). Compared to only personalized loss and anomalous loss, Maverick obtains an accuracy improvement, ranging from 0.8% to 3.7%. This result validates the effectiveness of both modules working in tandem. The ablation study demonstrates that all components are essential for effectively training clients' local models. Fig. 13 also illustrates the model convergence with different contrastive losses, showing that Maverick achieves the greatest speedup than only personalized loss and anomalous loss.

**Table 3: Ablation study of Maverick's top-1 accuracy with different contrastive losses on three datasets, where Pers., Anom., and Maverick denote the use of only personalized contrastive loss, only anomalous contrastive loss, and both losses, respectively.**

| Model | Dataset | Pers. | Anom. | Maverick |
|-------|---------|-------|-------|----------|
| LeNet-5 | Fashion-MNIST | 0.849 | 0.831 | 0.859 |
| ResNet34 | CIFAR-10 | 0.699 | 0.685 | 0.707 |
| | CINIC-10 | 0.693 | 0.676 | 0.709 |
| MobileNetV1 | CIFAR-10 | 0.633 | 0.628 | 0.649 |
| | CINIC-10 | 0.636 | 0.612 | 0.684 |

## 7 Related Work

**Federated Learning.** FL is an ML framework aimed at addressing privacy issues inherent in conventional cloud-based ML systems [54]. Clients in FL collaboratively aggregate a global model by exchanging local models, without exposing their private training data. FedAvg [43] is the pioneering aggregation method that averages clients' local models to produce a global model. Recently, a large amount of FL work has encompassed diverse aspects of FL, such as model convergence optimization [27, 36, 66], communication overhead reduction [41, 51], defense mechanisms to combat the poisoning attacks [14, 19, 26], privacy preservation for clients [50, 58], and model personalization [10, 20, 32, 53].

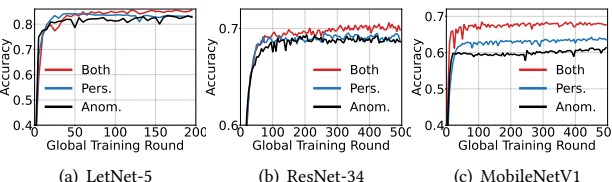

(a) LetNet-5      (b) ResNet-34      (c) MobileNetV1

**Figure 13: Model convergence with different contrastive losses across three datasets, i.e., only personalized loss (Pers.), only anomalous loss (Anom.), and both losses combined (Maverick): a) LetNet-5 on Fashion-MNIST; b) ResNet-34 on CIFAR-10; c) MobileNetV1 on CINIC-10.**

**Edge-assisted Federated Learning.** Traffic overhead is a significant challenge in FL due to the frequent model update transmissions between clients and the cloud server. Recently, edge computing starts to demonstrate its potential in supporting ML applications [46]. It enables edge-assisted FL systems that involve not only clients and the cloud but also edge servers [42]. In such a system, edge servers generate intermediate models by aggregating local models, and then transfer these models to the cloud server. The backhaul network traffic can be reduced immensely. Many studies have attempted to advance edge-assisted FL systems. Lim et al. [38] introduce a resource allocation mechanism where clients are treated as data owners, encouraging edge servers' participation. Wu et al. [60] propose HybridFL to improve training performance by selecting reliable clients. Feng et al. [15] investigate strategies for reducing the overheads during training associated with the transmission and aggregation of model parameters. Wang et al. [54] propose FedEdge to accelerate model training through performing local federated training (§2).

**Contrastive Learning.** Recently, self-supervised learning [40, 45, 65] focuses on learning effective data representations from unlabeled data. Among existing works, contrastive learning [8, 9] methods have attained state-of-the-art results in learning visual representations. It is employed by Maverick to mitigate model drifts in the edge-assisted FL system. Recently, Moon [34] pioneered the integration of contrastive learning into federated learning. Unlike traditional contrastive learning, Moon compares the representations learned by different models to accommodate clients' non-IID data.

## 8 Conclusion and Future Work

Edge-assisted federated learning (FL) systems are subject to model drifts under non-IID settings caused by imbalanced local federated training. To address this issue, this paper presented Maverick, a novel edge-assisted FL system that trains personalized local models for clients through contrastive local training. To further accelerate model convergence and accuracy, Maverick incorporates anomalous model-contrastive training into clients' contrastive local training, leveraging anomalous models as negative samples. Compared to state-of-the-art systems, Maverick demonstrates superior advantages in both model convergence and model accuracy. In the future, we will study model heterogeneity in edge-assisted FL and its impact on Maverick.

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

## A  Appendix

### A.1  Discussion

**Computational Overhead.** In Maverick, personalized and anomalous model-contrastive learning introduces additional computational overhead. For a given input $x$, clients need to compute its representations with the global model $W_G$, the intermediate model $W_I$, and anomalous models $W_A \in A$. Compared to local model training, this representation calculation process only requires forward propagation. In addition, during contrastive local training, clients only need to compute the representation of $W_G$ once for each global training round and the representation of $W_I$ and $W_A$ once for each local federated training round. In each local federated training round, clients usually conduct multiple local training epochs [43, 54, 56]. Thus, compared with the training overhead, the computational overhead incurred by model representation calculation is not significant.

Fig. 14 illustrates the numerical results. This experiment shows the ratios of extra and overall computational time on clients, where clients only conduct a single local training epoch during each local federated training round. As shown in Fig. 14, it is evident that even with a single local training epoch, the extra computation cost is negligible, averaging at only 0.39%.

**Storage Overhead.** Maverick also introduces additional storage overhead due to the inclusion of model parameters for $W_G$, $W_I$, and $W_A$, as well as $z_G$, $z_I$, and $z_A$. In ML model training, the main components of storage overhead include training data, model parameters, intermediate computation activations, optimizer states, and checkpoints. It is well known that storing activations of all intermediate layers for backpropagation consumes the majority of memory resources [6, 18]. From Fig. 14, we can see that even with a single local training epoch, the extra storage cost incurred by Maverick is negligible, averaging at only 0.68%.

**Communication Overhead.** In Maverick, personalized and anomalous model-contrastive learning introduces extra communication overhead because edge servers need to transmit extra models, i.e., the global model, and anomalous models to their clients. To avoid excessive communication overhead, Maverick employs a top-$k$ selection mechanism to cap a maximum of $k$ anomalous models for transmission to each client (§5.2).

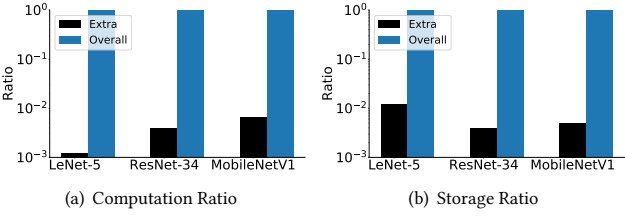

(a) Computation Ratio    (b) Storage Ratio

**Figure 14: The ratios of extra and overall computation time and storage costs in a single local epoch, where LeNet-5 [31] is trained Fashion-MNIST [61], ResNet-34 [21] is trained on CIFAR-10 [29] and MobileNetV1 [22] is trained on CINIC-10 [11] dataset.**

### A.2  Pseudocode

---

**Algorithm 1:** Training process of Maverick

---

/* clients train local models based on three losses, i.e., $l_{sup}$, $l_p$, and $l_a$, */

1 **Function** ClientTraining()
2   **For each** epoch $i = 1, 2, ...$
3    **For each** $(x, y) \in D^u$
    /* calculate supervised loss */
4     $\ell_{sup} \leftarrow CrossEntropyLoss(F_{W_L}(x), y)$
5     $z_L \leftarrow R_{W_L}(x)$
6     $z_I \leftarrow R_{W_I}(x)$
7     **For each** $A \in A$
8      $z_A \leftarrow R_{W_A}(x)$
    /* calculate personalized model-contrastive loss */
9     $\ell_p \leftarrow -\log \dfrac{\exp(sim(z_L, z_I)/\tau)}{\exp(sim(z_L, z_I)/\tau) + \exp(sim(z_L, z_G)/\tau)}$
    /* calculate anomalous model-contrastive loss */
10     $\ell_a \leftarrow$
     $- \sum_A \frac{d_A}{d_T} \log \dfrac{\exp(sim(z_L, z_I)/\tau)}{\exp(sim(z_L, z_I)/\tau) + \exp(sim(z_L, z_A)/\tau)}$
11     $\ell \leftarrow \ell_{sup} + \mu_p \ell_p + \mu_a \ell_a$
12     $W_L \leftarrow W_L - \eta \nabla \ell$
13   Send $W_L$ to its edge server

/* edge server aggregates local models to produce intermediate models */

14 **Function** EdgeAggregation()
15   **For each** $u = 1, 2, ...$
16    $W_L \leftarrow$ ClientTraining($W_G^{t-1}, W_I, A$)
17   categorize local models into anomalous ones ($A$) and genuine ones ($G$)
18   **If** receive new $W_G^t$ **then**
19    $W_I = \frac{1}{2}W_I + \frac{1}{2}W_G^t$
20    Send $W_I$ to the cloud server
21    Send $W_G^t$ to clients
22   **Else**
23    $W_I = \sum_{u \in G} \frac{1}{|G|} W_L$
24   Send $W_I$ and $A$ to clients.

/* cloud server aggregates intermediate models to produce global model */

25 **Function** CloudAggregation()
26   **For each** $t = 0, 1, ..., T-1$
27    **For each** $m = 1, 2, ..., M$ **in parallel**
28     $W_I^m \leftarrow$ EdgeAggregation()
29    $W_G \leftarrow \sum_{m=1}^{M} \frac{1}{M} W_I^m$
30    send $W_G$ to edge servers

---

