# OpenReview forum: "Maverick: Personalized Edge-Assisted Federated Learning with Contrastive Training"
_ACM.org/TheWebConf/2025/Conference — WWW 2025 Oral_

### Official Review · Reviewer_is9L · 2024-11-25

**Novelty:** 3
**Technical Quality:** 4

**Review:**

1. This paper incorporates the idea of contrastive learning into edge-assisted FL and conducts massive experiments to prove its effectiveness. But the core idea of this method is too similar to the paper Moon,  which seems to be a bit uninventive.
2. This paper uses anomalous models to construct negative samples, which makes effective use of poor-quality local models and has certain innovation. However, the rationality of this detection method is not proved. In addition, this paper does not consider the additional overhead that may be caused by this detection method in practical application.
3. This paper mainly uses "Top-1 accuracy" to prove that the proposed method "Maverick" can mitigate the model drifts. The diversity of the evaluation index is slightly insufficient. More evaluation index could be considered, such as "model heterogeneity" used in some papers.

**Questions:**

1. There exits grammatical error. The sentence "driving the training of clients’ models in the right direction away these anomalous models" on line 178 misses the word "from".
2. The assumption of anomalous models (served as negative samples) may not be satisfied in some scenarios. And the ablation study confirms the effectiveness of Anomalous Model-Contrastive Loss.
3. This paper uses Adaptive-Krum to categorize local models into genuine models and anomalous models, but lacks the validity proof of this method. What's the reason of choosing this approach, and what are its advantages over other partitioning methods? How does it affect the results of the experiment?
4. This paper uses the cosine similarity to calculate model similarity, which is also used in paper Moon to calculate the similarity of representation of images. Is it reasonable to take the same method as Moon to compute model similarity? Other than cosine similarity, are there more suitable methods for calculating model similarity?

**Reviewer Confidence:**

4: The reviewer is certain that the evaluation is correct and very familiar with the relevant literature

**Scope:**

3: The work is somewhat relevant to the Web and to the track, and is of narrow interest to a sub-community

---

### Official Review · Reviewer_Lnji · 2024-11-28

**Novelty:** 4
**Technical Quality:** 3

**Review:**

This paper presents Maverick, an edge-assisted federated learning system addressing model drifts via personalized local training with a model-contrastive loss. It includes anomalous models as negative samples to accelerate convergence. Experiments show Maverick improves convergence by up to 16.2x and accuracy by up to 12.7% over existing systems.

Pros:
- Introducing constractive training seems a novel idea to optimize federated learning, though the rationality is not well-explained.
- Methodologies are detailed and well-explained. The figures in the paper further improve the readability.
- The experiments are detailed, though some baselines need to be further included and some experimental settings are questionable.

Cons:
- My biggest concern about this paper is the rationality of considering constractive training. Although the authors demonstrate many figures in Section 3 (Motivation), they are used to show potential convergence performance problems in existing federated learning algorithms. The intuition of introducing constractive training is not described clearly. Besides, balancing local and global models have been considered for not a short time, which is usually called personalized federated learning. What are the advantages of constractive training compared with other strategies, such as regularization, Bayesian inference, and adaptive local aggregation?
- The authors claim in the first contribution that Maverick is the first edge-assisted FL system that mitigates the model drift issue caused by clients' imbalanced local training. This seems not the real situation. The authors may need to further clarify what "imbalanced local training" really refer to. As far as I know, neither system heterogeneity (straggler) nor data heterogeneity (non-IID) are new concepts in FL.
- The designed anomalous model-constrastive training strategy seems to highly rely on the Adaptive-Krum algorithm. Why does this paper consider this algorithm? According to the future work at the end of reference [54], which proposes Adaptive-Krum, a number of more sophisticated model poisoning attacks
have been proposed very recently. These attacks may escape Adaptive-Krum. Simply relying on this algorithm may not be effective anymore.
- The experiments in this paper seem to consider the situation with only one straggler. How about the situations with multiple stragglers, each of which has a different number of local federated learning rounds?

**Questions:**

- I am confused of the definition of straggler in this paper. Based on my understanding, a straggler usually uploads outdated models, or can conduct less number of local federated learning rounds than others in a single global round due to resource constraints. However, in the left column of page 2, the authors claim that straggler clients perform more local federated training rounds than others, which seems to contradict the traditional definition.
- The design of constrastive loss needs further discussion. What is the specific form for model representation?

**Reviewer Confidence:**

4: The reviewer is certain that the evaluation is correct and very familiar with the relevant literature

**Scope:**

4: The work is relevant to the Web and to the track, and is of broad interest to the community

---

### Official Review · Reviewer_KZNv · 2024-11-28

**Novelty:** 7
**Technical Quality:** 7

**Review:**

Quality

The paper is well-written and provides a detailed explanation of Maverick, the proposed edge-assisted federated learning (FL) system. It clearly outlines the problem of model drift caused by imbalanced local training in FL systems and presents Maverick as a novel solution. The technical details, including the introduction of personalized and anomalous model-contrastive losses, are robust and supported by comprehensive experiments.

Clarity

The paper is clear and structured logically, with sections that build on each other. The use of figures and tables enhances understanding, and the pseudocode and mathematical formulations are well-presented. However, some sections, such as the discussion of baseline comparisons, could benefit from additional elaboration to ensure accessibility to non-specialist readers.

Originality

Maverick introduces innovative techniques:

- Personalized Model-Contrastive Loss: Guides local models closer to intermediate models while diverging from the global model.
- Anomalous Model-Contrastive Loss: Uses anomalous models as negative samples to improve convergence and accuracy.

These approaches are distinct from existing FL systems and contribute new insights to the field.

Significance

The results demonstrate significant improvements in model accuracy (5.2%–12.7%) and convergence speed (1.4x–16.2x) compared to state-of-the-art methods. These findings underline the practical importance of the work in addressing model drift and accelerating FL system performance, particularly in edge computing scenarios.

Pros

Novelty: Introduction of contrastive losses to mitigate model drift is a meaningful contribution to FL research.

Experimental Rigor: Extensive evaluations on diverse datasets and models validate the effectiveness of Maverick.

Scalability: The focus on edge-assisted FL systems addresses a key challenge in deploying FL at scale.
Impact: The method shows improvements in both accuracy and speed, making it relevant for real-world applications.

Comprehensive Results: The paper compares Maverick against multiple baselines, demonstrating its superiority.

Cons

Complexity: The introduction of multiple contrastive losses may increase computational overhead and implementation complexity.

Limited Real-World Validation: Experiments are conducted in controlled environments; real-world deployment scenarios with heterogeneous devices and networks are less emphasized.

Focus on Anomalous Models: While innovative, the reliance on anomalous models as negative samples may require additional work to ensure robust detection and classification of anomalies.

**Questions:**

1. Clarification on Methodology

- How does Maverick handle highly imbalanced datasets where some clients contribute significantly less data than others? Does this impact the performance of the personalized and anomalous model-contrastive losses?

- Could you elaborate on how the anomalous models are identified using existing methods? How robust is this detection in scenarios with overlapping distributions between genuine and anomalous models?

2. Impact on Real-World Deployments
- The experiments are conducted in a controlled environment. Have you considered validating Maverick in real-world heterogeneous environments with varying network latencies, client hardware capabilities, or adversarial conditions?

- How would Maverick handle scalability if the number of clients increases significantly, particularly with regard to communication and computation overhead?

3. Computational Overhead

- While the results highlight Maverick's improvements in accuracy and convergence, how significant is the additional computational cost introduced by the personalized and anomalous model-contrastive losses?

- Would this overhead be prohibitive for resource-constrained edge devices?

5. Limitations
- Anomalous models are used as negative samples to guide convergence. However, could there be situations where these models introduce bias or negatively affect the global model? How does Maverick mitigate such risks?

- How sensitive is Maverick's performance to the hyperparameters ($\mu_p$ and $\mu_a$)? Did you observe consistent performance across datasets, or are these parameters highly task-dependent?

**Reviewer Confidence:**

1: The reviewer's evaluation is an educated guess

**Scope:**

4: The work is relevant to the Web and to the track, and is of broad interest to the community

---

### Official Review · Reviewer_6cVn · 2024-12-01

**Novelty:** 4
**Technical Quality:** 3

**Review:**

**Summary**

The paper introduces Maverick, a novel edge-assisted federated learning system designed to mitigate model drifts caused by imbalanced local training in non-IID settings. Maverick employs contrastive learning to train personalized local models for clients, aligning their models with corresponding intermediate models rather than a global model. It also incorporates anomalous model-contrastive training, using poor-quality models as negative samples to guide model convergence. The system includes a top-k selection mechanism to manage communication overhead. Extensive experiments demonstrate that Maverick outperforms state-of-the-art edge-assisted FL systems in terms of model accuracy and convergence speed.

**pros**

1.Maverick is the first edge-assisted FL system that mitigates the model drift issue caused by clients’ imbalanced local training.

2.Maverick significantly improves model accuracy and convergence speed.

**cons**

1.Figure 7(b) seems to be inconsistent with the author's description.

2.Although the author has presented the convergence of the models trained in different systems through experiments, I believe that including a theoretical analysis of Maverick's convergence could enhance the theoretical contribution of the paper.

3.For small datasets such as CIFAR-10, I think multiple experiments could be conducted and the mean and std reported to eliminate the impact of noise on the experimental results.

**Questions:**

Consistent with the points mentioned above in the cons.

**Reviewer Confidence:**

2: The reviewer is willing to defend the evaluation, but it is likely that the reviewer did not understand parts of the paper

**Scope:**

2: The connection to the Web is incidental, e.g., use of Web data or API

---

### Official Review · Reviewer_Pxwg · 2024-12-04

**Novelty:** 6
**Technical Quality:** 6

**Review:**

The paper proposes the framework Maverick which uses the anomalous models to assist in the personalised contrastive learning to manage the model-drift. The overall idea is interesting and the paper is well-written. The authors have included multiple baseline comparisons and detailed results to highlight the performance of Maverick.

Some of the comments regarding the draft are:
1. I would suggest that the authors briefly describe Adaptive-Krum in the paper as that serves as the key technique to categorise the models into genuine and anomalous.

2. What is the communication cost of Maverick? I see that authors have discussed this multiple times in the draft (as well as in the appendix) but a proper quantification of the cost and a fair comparison with the baselines considering different numbers of anomalous models (k) is necessary.

3. I see that the authors have performed have considering 50 clients. It would be interesting to see if the number of clients is further scaled up then whether the communication overhead can become a significant bottleneck for Maverick.

**Questions:**

Please refer to the main review.

**Reviewer Confidence:**

4: The reviewer is certain that the evaluation is correct and very familiar with the relevant literature

**Scope:**

4: The work is relevant to the Web and to the track, and is of broad interest to the community